# Predicting development of adolescent drinking behaviour from whole brain structure at 14 years of age

Simone Kühn[1,2]*, Anna Mascharek[1], Tobias Banaschewski[3], Arun Bodke[4], Uli Bromberg[5], Christian Büchel[5], Erin Burke Quinlan[6], Sylvane Desrivieres[6], Herta Flor[7,8], Antoine Grigis[9], Hugh Garavan[10,11], Penny A Gowland[12], Andreas Heinz[13], Bernd Ittermann[14], Jean-Luc Martinot[15], Frauke Nees[3,8], Dimitri Papadopoulos Orfanos[9], Tomas Paus[16,17,18], Luise Poustka[19], Sabina Millenet[3], Juliane H Fröhner[20], Michael N Smolka[20], Henrik Walter[13], Robert Whelan[21], Gunter Schumann[6], Ulman Lindenberger[2], Jürgen Gallinat[1], IMAGEN Consortium

[1]Department of Psychiatry and Psychotherapy, University Medical Center Hamburg-Eppendorf, Hamburg, Germany; [2]Center for Lifespan Psychology, Max Planck Institute for Human Development, Berlin, Germany; [3]Department of Child and Adolescent Psychiatry and Psychotherapy, Central Institute of Mental Health, Medical Faculty Mannheim, Heidelberg University, Mannheim, Germany; [4]Discipline of Psychiatry, School of Medicine and Trinity College Institute of Neuroscience, Trinity College Dublin, Dublin, Ireland; [5]University Medical Centre Hamburg-Eppendorf, Hamburg, Germany; [6]Medical Research Council - Social, Genetic and Developmental Psychiatry Centre, Institute of Psychiatry, Psychology & Neuroscience, King's College London, London, United Kingdom; [7]Department of Cognitive and Clinical Neuroscience, Central Institute of Mental Health, Medical Faculty Mannheim, Heidelberg University, Mannheim, Germany; [8]Department of Psychology, School of Social Sciences, University of Mannheim, Mannheim, Germany; [9]NeuroSpin, CEA, Université Paris-Saclay, Gif-sur-Yvette, France; [10]Department of Psychiatry, University of Vermont, Burlington, United States; [11]Department of Psychology, University of Vermont, Burlington, United States; [12]Sir Peter Mansfield Imaging Centre School of Physics and Astronomy, University of Nottingham, Nottingham, United Kingdom; [13]Department of Psychiatry and Psychotherapy, Charité – Universitätsmedizin Berlin, Berlin, Germany; [14]Physikalisch-Technische Bundesanstalt (PTB), Berlin, Germany; [15]Institut National de la Santé et de la Recherche Médicale, INSERM Unit 1000 "Neuroimaging & Psychiatry", University ParisSud, University Paris Descartes, Paris, France; [16]Bloorview Research Institute, Holland Bloorview Kids Rehabilitation Hospital, Toronto, Canada; [17]Department of Psychology, University of Toronto, Toronto, Canada; [18]Department of Psychiatry, University of Toronto, Toronto, Canada; [19]Department of Child and Adolescent Psychiatry and Psychotherapy, University Medical Centre Göttingen, Göttingen, Germany; [20]Neuroimaging Center,Department of Psychiatry, Technische Universität Dresden, Dresden, Germany; [21]Global Brain Health Institute,School of Psychology, Trinity College Dublin, Dublin, Ireland

*For correspondence: kuehn@mpib-berlin.mpg.de

**Abstract** Adolescence is a common time for initiation of alcohol use and development of alcohol use disorders. The present study investigates neuroanatomical predictors for trajectories of future alcohol use based on a novel voxel-wise whole-brain structural equation modeling framework. In 1814 healthy adolescents of the IMAGEN sample, the Alcohol Use Disorder Identification Test (AUDIT) was acquired at three measurement occasions across five years. Based on a two-part latent growth curve model, we conducted whole-brain analyses on structural MRI data at age 14, predicting change in alcohol use score over time. Higher grey-matter volumes in the caudate nucleus and the left cerebellum at age 14 years were predictive of stronger increase in alcohol use score over 5 years. The study is the first to demonstrate the feasibility of running separate voxel-wise structural equation models thereby opening new avenues for data analysis in brain imaging.
DOI: https://doi.org/10.7554/eLife.44056.001

## Introduction

Adolescence is a critically vulnerable time for the development of alcohol drinking habits that may lead to considerable consequences later in life including the development of alcohol addiction. Importantly, the period of adolescence coincides with substantial behavioural changes together with structural and functional brain development. Cortico-striatal regions play an important role in the regulation of behaviour and might therefore play a role in progress and maintenance of habits such as drinking (*Heinz, 2002*). In particular, it has been proposed that drug addiction involves dysfunctions of brain circuitry related to the neurotransmitter dopamine that lead to alterations in both impulsive and compulsive behaviour (*Koob and Kreek, 2007*). Since early exposure to drugs may alter brain development during adolescence this may set the stage for cognitive problems in adulthood, which translate into behavioral consequences throughout life (*Jacobus and Tapert, 2013*). Hence, it is of particular importance to predict the acceleration of alcohol use as early as possible during adolescence in order to intervene timely.

Nees and colleagues reported that reward-related brain activation aided in the prediction of early-onset drinking in adolescents at age 14 years in a subset of the data used in the present paper (*Nees et al., 2012*). In a functional imaging study on youth of age 12–14 years, prior to initiation of alcohol use, it was found that teens classified as transitioning to heavy alcohol use by age 18 had less blood oxygen level dependent (BOLD) activation in frontal, temporal, and parietal cortices in a response inhibition task (*Norman et al., 2011*). These reports support the notion that brain differences present early during adolescence may leave certain youth vulnerable to addictive behaviors. In an earlier publication on the same data set, we focussed on the prediction of changes in alcohol-related problems between age 14 and 16 years based on gyrification of the orbitofrontal cortex (*Kühn et al., 2016*). We argued that it is important to use behavioral difference scores when aiming at predicting prospective behavior rather than absolute measures at the prospective time point only. In applying difference scores rather than absolute measures, all available information can be taken into account to evaluate potentially problematic behaviour.

Within the present study, we want to go beyond previous prediction attempts by including a trajectory of alcohol-related behavior over a longer period of time, from age 14 to 16 and 19 years of age. Obtaining three measurement occasions enables the modelling of change on a latent level within a structural equation modelling (SEM) framework. We set out to investigate brain structural predictors at age 14 of the latent change trajectory of the alcohol use score on a voxel-based whole-brain basis. Methodologically this goes beyond previous studies with an SEM approach including brain data, since commonly only extracted regions of interests or global brain measures (such as intracranial volume, white matter hyperintensities) have been employed in an SEM context (*Ritchie et al., 2015*; *Kievit et al., 2014*). The innovation of this paper's approach lies in analysing the whole brain on voxel level. In doing so, we are able to on the one hand extract possible predictors for change in alcohol consumption on a fine-grade level (i.e. voxel), and on the other hand to exploratory analyse the whole brain in search of predictive regions. This approach offers the major advantage of giving consideration to both, specific structures of the brain on a rather fine-graded level, and the broad, exploratory search for generally influential areas.

**eLife digest** Puberty is a time of transformation. Physical changes in the body occur alongside changes in personality and behaviour. Compared to children, adolescents tend to be risk-takers and novelty-seekers. They crave new sensations and experiences, as well as social interaction with their peers. It is around puberty that many people try alcohol for the first time. But it is not clear why people differ in their drinking habits, and why a small minority of young adults go on to become dependent on alcohol.

Part of the answer may lie in changes in the brain. Differences in the size and structure of brain regions contribute to differences in behaviour between individuals. During adolescence, the brain undergoes extensive re-modelling. It forms new connections, while also pruning away connections that are unused. Could differences in brain structure at puberty lead to differences in alcohol consumption in early adulthood?

Kühn et al. scanned the brains of about 1,800 healthy adolescents at the age of 14 and then again at 19 (within the context of the IMAGEN study). At three time points, the teenagers also filled in questionnaires about their use of alcohol. Two areas of the brain – the caudate nucleus and the left cerebellum – were larger at age 14 in teenagers who would increase their alcohol consumption by age 19. The larger the areas at age 14, the bigger the increase in alcohol consumption over time. Notably, there was no relationship between the size of either brain area at the age of 14 and how much alcohol the individuals drank at the same age.

These results may help us to understand why some young adults develop harmful drinking habits, whereas most do not. The findings are part of a large and complex picture. Other factors, such as social influences, also shape alcohol consumption. However, the findings of Kühn et al. suggest that differences in brain structure may make some individuals more likely to increase how much alcohol they drink than others. Understanding these biological differences could help researchers to develop measures to prevent addiction in young adults.

DOI: https://doi.org/10.7554/eLife.44056.002

## Results

We started the analyses with estimating the two-part latent growth curve model on the clinical data only containing an intercept and a linear growth factor for both alcohol use vs. non-use as well as for the alcohol use score. *Table 1* shows the classification of the AUDIT-scores concerning severity of use (*Figure 1*).

In the discrete part of the model, a linear growth model demonstrated better fit to alcohol use vs. non-use than an intercept-only model, $\Delta\chi^2$ (*Koob and Kreek, 2007*) (3, $N$ = 1814)=653.63, p<0.001. Inclusion of a quadratic growth factor did not improve model fit, $\Delta\chi^2$ (*Koob and Kreek, 2007*) (1, $N$ = 1814)=−0.74, p=0.390, therefore we refrained from a quadratic growth factor.

**Table 1.** Severity of alcohol use at three measurement occasions according to AUDIT.*

|  | Baseline n** = 1794 (100%) | Follow-up 1 n = 1439 (100%) | Follow-up 2 n = 1284 (100%) |
| --- | --- | --- | --- |
| No use at all | 855 (47.7%) | 255 (17.7%) | 95 (7.4%) |
| Unproblematic use | 872 (48.7%) | 961 (66.7%) | 823 (64.1%) |
| Medium level of alcohol problems | 64 (3.6%) | 218 (15.3%) | 329 (10.7%) |
| High level of alcohol problems | 2 (0.1%) | 4 (0.3%) | 28 (2.2%) |
| Indicating dependence | 1 (0.1%) | 1 (0.1%) | 9 (0.9%) |

*Note: Categorization is based on the interpretation guideline of the World Health Organization: Cut-offs scores are: 0–7 = unproblematic use, 8–15: simple advice focused on the reduction of hazardous drinking warranted, 16–19: brief counseling and continued monitoring warranted, above 20: further diagnostic for alcohol dependence strongly warranted.

**Note: 20 individuals had missing data, in total adding up to 1814.

DOI: https://doi.org/10.7554/eLife.44056.004

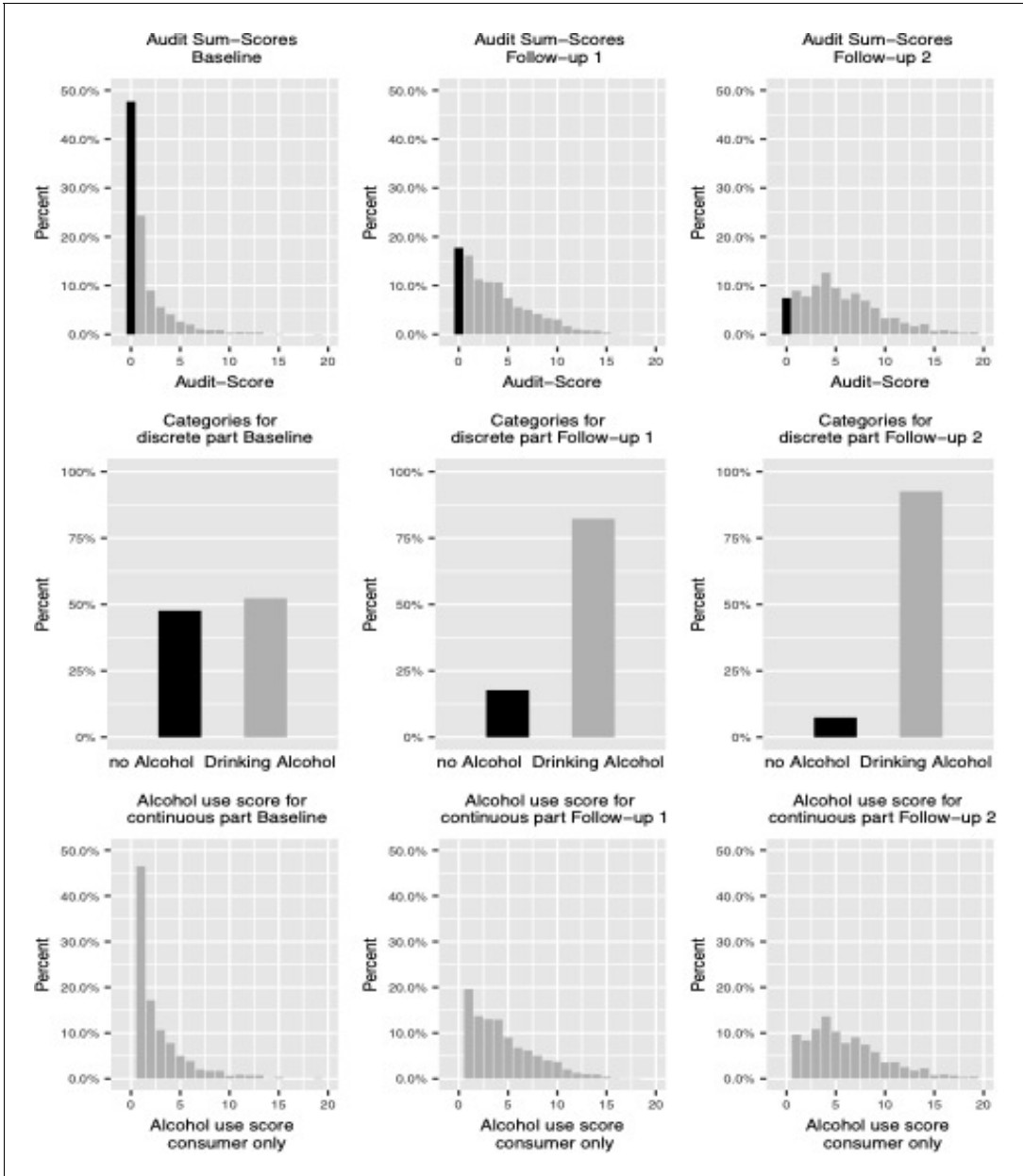

**Figure 1.** Preparation of AUDIT Sum-Scores for two-part latent growth mixture model. Upper row: data from original scale (Sum Score), zeros are shown in black and indicate non-drinking individuals. Middle row: Transformation of data into consumer and non-consumer without fine-grading of alcohol use scores. Bottom row: Alcohol use score (AUDIT Sum-Score) for individuals who drink at all. Note that to enhance readability of the figure, sum-scales (upper and bottom row) are truncated at a score of 20.
DOI: https://doi.org/10.7554/eLife.44056.003

In the continuous part of the model, a linear growth model likewise demonstrated a better fit to alcohol use scores than an intercept-only model, $\Delta\chi^2$ (***Koob and Kreek, 2007***)(3, N = 1814)=860.58, p<0.001. Inclusion of a quadratic growth factor did not improve model fit, $\Delta\chi^2$ (***Koob and Kreek, 2007***)(1, N = 1814)=−0.58, p=0.450. Therefore, we accepted the model with intercept and a linear growth factor on the clinical data as our working model. We then added the nuisance covariates age, sex, scanner and total brain volume to the model by predicting the slope of the continuous part of the model (no matter whether the regression paths were significant or not), since it is common practice in neuroimaging studies to control for these nuisance variables. The final model on the

clinical data, not yet including the brain data (since this varied for each voxel of the brain) demonstrated an acceptable model fit, $\chi^2 = 444$, df = 65, RMSEA = 0.057, CFI = 0.785, SRMR = 0.065.

Concerning the latent intercept and slope for both parts of the model, intercepts were significantly different from zero and variances of the intercepts for both parts of the model were significant, suggesting significant interindividual heterogeneity around the estimated mean level of alcohol drinking use vs. non-use and the alcohol use score at age 14 (for estimates see *Table 2*). The covariance between the intercepts of the two parts of the model was 0.124, p<0.001, indicating that adolescents with a higher propensity to engage in alcohol drinking also engaged in it more frequently and vice versa.

Turning to the growth parameters, for the continuous part of the model the estimated mean of the slope was not significantly different from zero, indicating that, on average, no change in drinking habits emerged over time. However, the variance was significantly different from zero, indicating interindividual differences in change of drinking behaviour between participants. For the discrete part of the model both the mean and the variance of the slope were significantly different from zero, indicating change on average as well as interindividually. The positive mean of the slope indicated an increasing propensity for drinking across time. Intercept and slope covaried significantly (−0.033, p<0.001) within the discrete part of the model, however, numerically the correlation coefficient was so small that we do not think the association necessitates in-depth interpretation. No significant correlation emerged between intercept and slope for the continuous part of the model, indicating that a relation between alcohol use score at age 14 and change in this behavior was not captured in our model.

**Table 2.** Estimated parameters in probability of use vs. non-use and alcohol use score with nuisance variables on the clinical data (not yet including brain data)

| | Intercept | | Slope | |
|---|---|---|---|---|
| | Estimate | SE | Estimate | SE |
| Part 1: Prevalence of alcohol drinking (use vs. non-use)=discrete part of the model | | | | |
| Mean | 0.568** | 0.011 | 0.188** | 0.006 |
| Variance | 0.090** | 0.009 | 0.024** | 0.004 |
| Part 2: Alcohol use score of AUDIT = continuous part of the model | | | | |
| Mean | 0.693** | 0.037 | 0.498 | 0.642 |
| Variance | 0.618** | 0.087 | 0.218** | 0.046 |
| Regression onto Part two slope | | | | |
| Sex | | | −0.183** | 0.046 |
| Age | | | −0.000 | 0.000 |
| TBV | | | 0.000* | 0.000 |
| Site_London | | | 0.410* | 0.163 |
| Site_Nottingham | | | 0.368* | 0.161 |
| Site_Dublin | | | 0.517* | 0.167 |
| Site_Berlin | | | 0.091 | 0.170 |
| Site_Hamburg | | | 0.122 | 0.162 |
| Site_Mannheim | | | 0.038 | 0.163 |
| Site_Paris | | | 0.079 | 0.163 |
| Site_Dresden | | | −0.044 | 0.163 |
| Covariances | | | | |
| Covariance between intercept and slope in Part 1 | −0.033** | 0.005 | | |
| Covariance between intercept and slope in Part 2 | −0.078 | 0.050 | | |
| Covariance between the intercepts of Part 1 and Part 2 | 0.124** | 0.012 | | |

*p < 0.05 , **p<0.001, SE = standard error, TBV = total brain volume

DOI: https://doi.org/10.7554/eLife.44056.005

Based on this two-part latent growth curve model on the clinical data, we entered the brain data and conducted a whole-brain analysis on grey matter probability maps at age 14 years predicting change in alcohol use score, that is the latent slope in the continuous part of the model, over time from each voxel in the brain (*Figure 2*). Within the model an increase in alcohol use score is reflected as a positive and a decrease in alcohol use score as a negative latent slope mean. We found a positive association within bilateral caudate nucleus (around −14, 24, 7 and 17, 24, 6) and left cerebellum (Lobule VIII/IX, around −16,−53, −56), where higher grey matter volume predicts greater change in alcohol use scores (p<0.001, cluster >100 voxels, *Figure 3*). We repeated the same analysis on log-transformed data, to address the problem of skewness in the data with almost identical results. The same analysis on white matter probability maps did not result in any significant clusters. Neither did repeating the same analyses when investigating a regression path between white or grey matter voxels and the change in alcohol use vs. non-use. Moreover, we conducted the same whole-brain analysis while predicting the latent intercept of the continuous part of the model, which reflects how high individuals score on AUDIT on average. Here also no significant clusters emerged, neither for white nor grey matter maps.

## Discussion

The goal of the present study was to unravel structural brain predictors at age 14 of the trajectory of alcohol use scores over the course of adolescence, namely between the age of 14 and 19 years. For this reason, we used a two-part latent growth curve model since it decomposes the semicontinuous outcome measure into a dichotomous use vs. non-use and a continuous alcohol use score part. The mean and variance of the intercepts for both parts of the model were significant, and the covariance between both intercepts was significant, indicating that adolescents with a higher propensity to engage in alcohol drinking also engaged in it more frequently and vice versa. For the continuous part of the model, the estimated mean of the slope was not significantly different, indicating that, on average, no change in drinking habits emerged. However, the variance was significant, indicating interindividual differences in change of drinking behaviour between participants. For the discrete part of the model, both the mean and the variance of the slope were significantly different from zero.

Since we were most interested in early brain-predictors of the intraindividual changes of alcohol use scores we computed a separate SEM for each brain voxel acquired at baseline (age 14 years). To obtain a brain map we plotted the resulting statistics of the regression path between brain and changes in alcohol use score back into brain space where we observed that higher grey matter volume in bilateral caudate nucleus and in left cerebellum was associated with a stronger increase (slope) in alcohol use scores. No associations were observed between grey matter brain data and the slope or intercept of the dichotomous use vs. non-use or the intercept of the continuous part of the model, nor in the white matter of the brain.

### Neurodevelopmental changes in caudate and cerebellum

The direction of the association, namely between increase in alcohol use scores and higher brain volume is remarkable, since it may reflect a disturbance of brain development during adolescence. Caudate and cerebellum have been described as undergoing changes over the lifetime that resemble an inverted U-shape in gray matter volume that peaks during adolescence (*Durston et al., 2001*; *Brenhouse and Andersen, 2011*). However, on a longitudinal data set the changes of caudate over the course of adolescence were the smallest in comparison to other subcortical brain structures with caudate, putamen and nucleus accumbens peaking at earlier ages than amygdala and hippocampus (*Goddings et al., 2014*). In tendency females seem to show a peak around age 11 years and males seem to show a decrease in caudate volume over adolescence (*Goddings et al., 2014*; *Brain Development Cooperative Group, 2012*). The pattern of changes in the cerebellum seem less clear with one study reporting decreases across adolescence (*Ostby et al., 2009*) and another showing an inverse U-shape pattern with a peak around 15 years of age (*Brain Development Cooperative Group, 2012*). However, the literature on age related changes in brain volume during adolescence does not help to solve the question whether our observed effects reflect a deceleration of pruning in bilateral caudate and cerebellum or an overproduction that surpasses the normal overproduction of synapses.

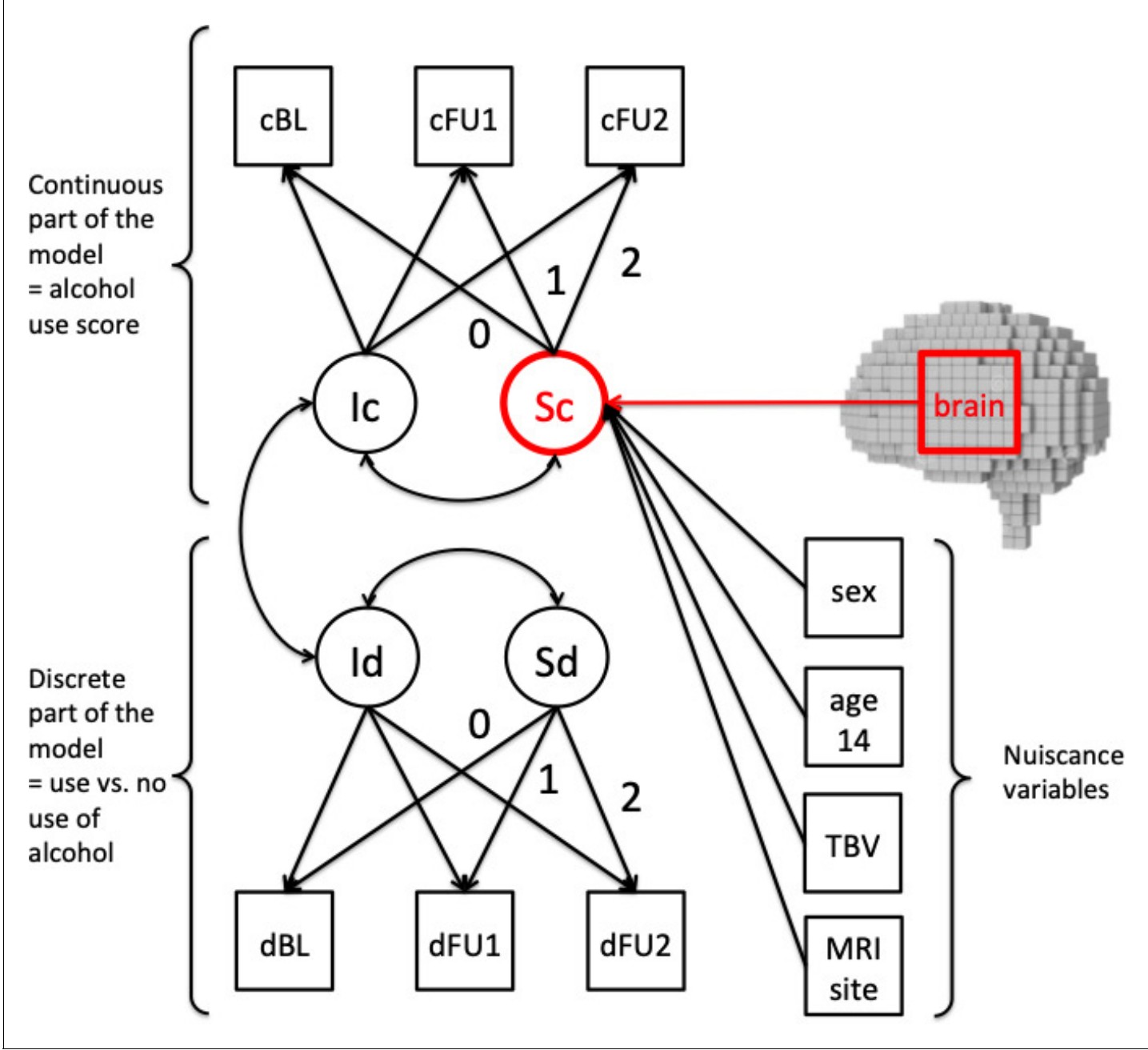

**Figure 2.** Two-part latent growth mixture model.  c = continuous, d = discrete, BL = baseline, FU = follow up, I = intercept, S = slope, TBV = total brain volume, MRI site was not a single indicator as depicted for reasons of simplicity, but consisted of 9–1 separate indicators dummy coding the different scanners used.
DOI: https://doi.org/10.7554/eLife.44056.006

### Caudate nucleus and cerebellum as a predictor of alcohol problems

The present finding of a predictive value of bilateral caudate volume for the trajectory of alcohol use during adolescence fits nicely to previous studies in search of brain structural predictors of drinking.

One study on 40 adolescents using FreeSurfer showed that at baseline (12–17 years of age) participants who transitioned into heavy drinking after 3 years showed smaller left cingulate, pars triangularis, and rostral anterior cingulate volume, and less right cerebellar white matter volumes (*Squeglia et al., 2014*).

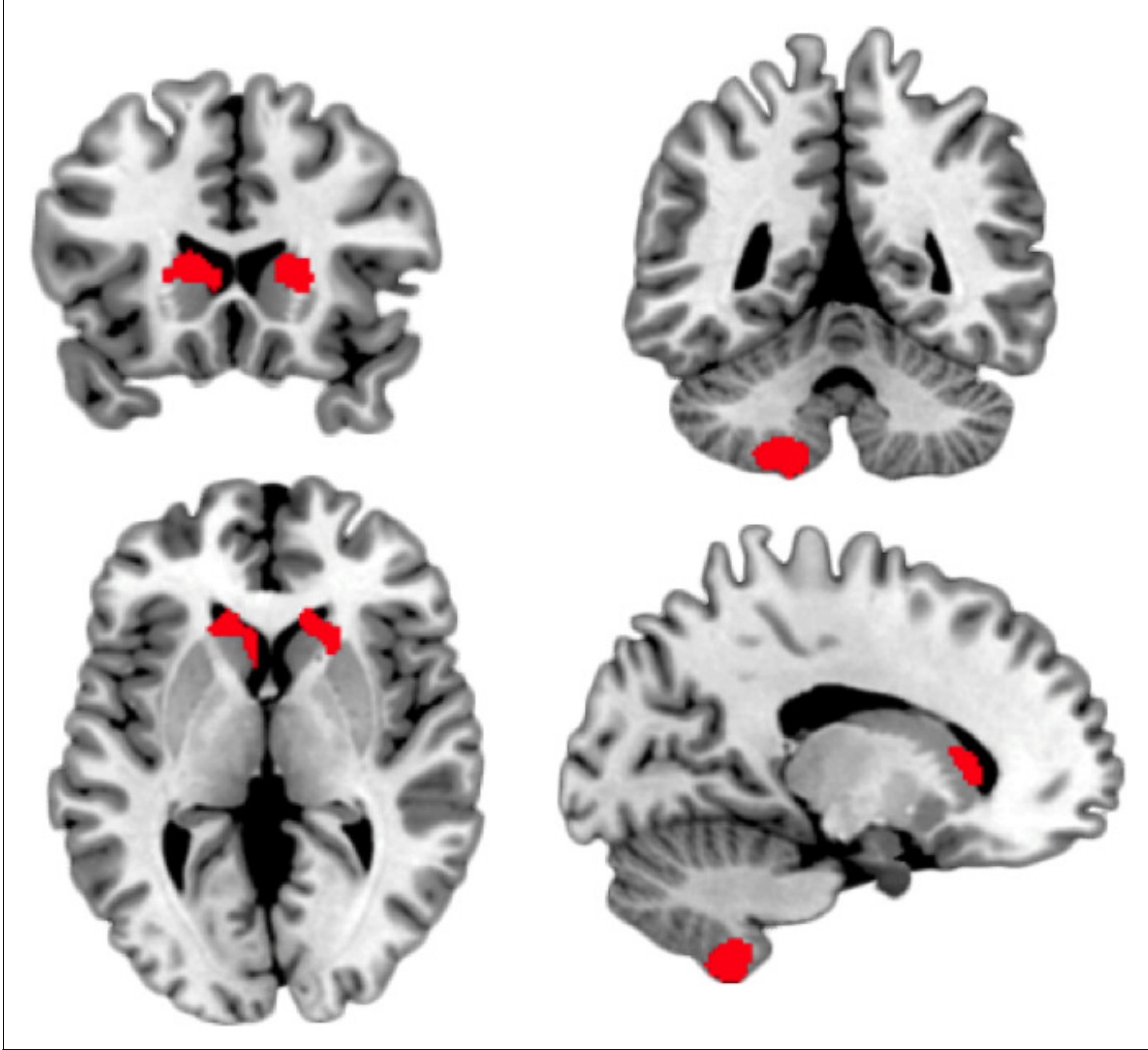

**Figure 3.** Brain regions showing a significant regression path from brain voxel to the latent slope of alcohol use score increase over time. The higher the grey matter volume the larger the slope increase.

DOI: https://doi.org/10.7554/eLife.44056.007

Other studies focussed on activation of the striatum, for example during an fMRI reward paradigm, reporting increases in brain activation in caudate nucleus at reward delivery in participants who had their first drink during puberty as compared to those who started after puberty (*Boecker-Schlier et al., 2017*). In another study focussing on brain function, 43 postpubertal participants (aged 18–21 years) were followed over a year and classified into moderate and heavy drinkers or transitioners who started drinking heavily. In a cue-reactivity paradigm showing alcohol stimuli, transitioners showed higher activation in bilateral caudate nucleus, orbitofrontal cortex medial frontal cortex/anterior cingulate and left insula (*Dager et al., 2014*). However, the two latter results have not been corrected for grey matter volume, and it could well be that the stronger brain activation

observed in caudate (among others) was actually driven by higher grey matter volume (to correct for these structural effects dedicated toolboxes have been developed; *Casanova et al., 2007*).

Of note, it is very interesting that the cluster in caudate nucleus and cerebellum found to be predictive of the future changes in alcohol use scores were not related to the mean level of alcohol use scores in each individual, since we observe no significant regions in the analysis where the latent intercept of the continuous part of the model predicts the respective brain voxels. This indicates that the grey matter volume in caudate nucleus and cerebellum has a value in predicting changes, but not the average alcohol use in general at the timepoint analysed. On the basis of the present analyses, one might argue that preventive measures at this age should focus on the development of alcohol use rather than on drinking habits at the age of 14, as the results suggest that specific characteristics of those brain regions prepare the ground for future alcohol consumption.

## Prediction of the start of alcohol drinking

The fact that the brain-based prediction of the latent slope of the dichotomous part of the model showed no significant resulting clusters indicates that, using the presented methods, brain data is *not* indicative for the prognosis when individuals start drinking. In this sense, the present results could be regarded as in line with a previous publication from the IMAGEN data set (*Whelan et al., 2014*) in which it was shown that brain information (at age 14 years) added only moderately in the prediction of binge drinking at age 16 years. Moreover, rather unspecific brain measures have been added in the analyses, namely overall regional grey matter volume and the ratio of grey and white matter volume. In a different data set (*Squeglia et al., 2017*), likewise only diffuse regions added to the prediction of initiation of alcohol use during adolescence. This later observation makes the prediction of the changes in alcohol use scores over time from very distinct brain regions the more remarkable. While we learned from previous studies that brain characteristics might not be the key factor in explaining the start of alcohol consumption in adolescence, we learn from the present study that brain structural characteristics are of relevance when considering the development of alcohol use in adolescence.

## Caudate nucleus and psychiatric disease

Interestingly, a recent meta-analysis on brain imaging studies focussing on brain structural alterations across psychiatric disorders has revealed consistency in increases within bilateral striatum when comparing psychiatric patients to controls (*Goodkind et al., 2015*). From this meta-analysis on cross-sectional data, it is unclear whether these striatal increases in psychiatric patients were already present during adolescence or whether they occurred around disease onset or over the course of the disease. But it is interesting that the direction of the effect and the localisation bear resemblance to the results of our study, although we focussed on trajectories of alcohol use which were, for most of the participants, far from the actual diagnosis of alcohol addiction.

## Novel methodological approach: whole-brain structural equation modeling

To our knowledge up to now structural equation modeling on brain imaging data has been conducted solely based on data derived from regions of interest (ROIs) (*Kievit et al., 2014*; *Kühn et al., 2017*; *McArdle et al., 2004*; *Raz et al., 2005*). However, this has the strong disadvantage that the obtained results are restricted to the regions selected for the analysis at hand. The present study demonstrates the feasibility of running separate structural equation models for each and every voxel of the brain and therefore plot the voxel-wise results of structural equation models back into brain space. This approach is not restricted to growth curve models but can be applied to all models in an SEM context. It can for example also be applied to measurement models, in order to relate brain structure or function not only to a separate task performance measure but rather to the latent factor representing the shared variance of a set of different performance measures from the same cognitive domain. This offers a new avenue of structural equation modeling on neuroimaging data in an unbiased, comprehensive way.

## Conclusion

The present study revealed structural brain predictors (at 14 years of age) of the trajectory of alcohol use scores between the age of 14 and 19 years. A two-part latent growth curve model was utilized to decompose the semicontinuous AUDIT outcome measure into a dichotomous use vs. non-use and a continuous alcohol use scale part. We predicted the slope of use vs. non-use and of alcohol use scores by voxel-wise grey and white matter probability maps at baseline. To obtain brain maps as a result, we plotted the statistics of the regression path between brain and slope back into brain space. We observed that higher grey matter volume in bilateral caudate nucleus and in left cerebellum at age 14 years was associated with a stronger increase in alcohol use scores. This finding fits well to previous studies pointing at an association between increases in striatum and psychiatric disease. Potentially this is due to neurodevelopmental interindividual differences since adolescence is a period of brain structural in- and decreases. Our finding may reflect a deceleration of pruning or an overproduction that surpasses the normal developmental overproduction of synapses. Future research with repeated neuroimaging measurements is needed to solve this neurodevelopmental question.

# Materials and methods

## Participants

We used data of 1794 healthy 14-year-old adolescents (mean age = 14.4, SD = 0.45 years; 54% males) who were recruited within the scope of the IMAGEN project, a European multi-centre genetic-neuroimaging study in adolescence (*Schumann et al., 2010*). The selection of the participants was based on the fact that structural imaging data at age of 14 years was present. At the time of analyses reported here (age 14 years), retest data at age 16–17 years was present for 1439 participants (mean age = 16.6, SD = 0.64 years; 55% males) and at age 19 years for 1284 participants (mean age = 19.0, SD = 0.77 years; 53% males). Written informed consent was obtained from all participants as well as from their legal guardians. The adolescents were recruited from secondary schools. The study was approved by all local ethics committees separately (in Germany this was accomplished by the medical ethics committee of the University of Heidelberg, reference number: 2007-024N-MA) and approved by the head teachers of the respective schools. Participants with a medical condition or neurological disorders were excluded. All participating subjects were assessed by means of self-rating and two external ratings (by their parents and a psychiatrist specialized in pediatrics) based on ICD-10 as well as DSM-IV (The Development and Well-Being Assessment Interview, DAWBA; *Goodman et al., 2000*).

## Questionnaire

We administered the Alcohol Use Disorder Identification Test (AUDIT, *Babor and Higgins-Biddle, 2001*) at Baseline (age 14 years), Follow-up 1 (age 16–17 years) and Follow-up 2 (age 19 years) to identify alcohol use. We computed the total score by adding the scores of all 10 items.

## Scanning procedure

Structural MRI was performed on 3 Tesla scanners from three manufacturers (Siemens: five sites; Philips: two sites; and General Electric: two sites). The details of the entire MR protocol are described elsewhere (*Schumann et al., 2010*). In this study, we used the T1-weighted images. These high-resolution anatomical MRIs were obtained using a three-dimensional magnetization prepared gradient-echo (MPRAGE) sequence based on the ADNI protocol (http://adni.loni.usc.edu/methods/documents/mri-protocols/; modified for the IMAGEN study to give a $1.1 \times 1.1 \times 1.1$ mm$^3$ voxel size).

## Voxel-based morphometry

For the present report, structural MR data of 2072 adolescents were available. We excluded all participants where the image quality was suboptimal, most likely due to movement. The visual quality control was carried out by 10 independent raters.

Structural data acquired at age 14 years was preprocessed by means of the VBM8 toolbox (http://dbm.neuro.uni-jena.de/vbm.html) and SPM8 (http://www.fil.ion.ucl.ac.uk/spm) with default parameters. The VBM8 toolbox involves bias correction, tissue classification and affine registration.

The affine registered grey matter (GM) and white matter (WM) segmentations were used to build a customized DARTEL (diffeomorphic anatomical registration through exponentiated lie algebra) template. Then warped GM and WM segments were created. Modulation was applied in order to preserve the volume of a particular tissue within a voxel by multiplying voxel values in the segmented images by the Jacobian determinants derived from the spatial normalization step. In effect, the analysis of modulated data tests for regional differences in the absolute amount (volume) of GM/WM. Images were smoothed with a FWHM (full-width at half maximum) kernel of 8 mm.

## Structural equation modelling

Analyses were conducted within a structural equation modeling (SEM) framework using MPlus and R. We implemented a two-part latent growth curve model (*Muthen, 2001*; *Olsen and Schafer, 2001*) since the AUDIT scores were zero inflated. In a two-part latent growth curve model, zeros are valid values with its own meaning and not just proxies for missingness. Information that is contained by zeros and the specific values of the non-zeros is qualitatively different and might even be differentially influenced by covariates (cf., *Olsen and Schafer, 2001*). In a two-part latent growth curve model, the presence or absence of a behavior and, if present, the manifestation of a specific behavior can be modeled simultaneously in one model. In our study, the original distribution of the alcohol use variable (AUDIT-score) was decomposed into two parts (see *Figure 1*). Then, each was modeled by separate, but correlated, growth functions (see *Figure 2*). For the discrete part of the model scores of zero were separated from the rest of the distribution by creating a binary indicator variable that distinguished any positive alcohol use score (=1) from nouse (=0) (lower part of *Figure 2*). For the continuous part of the model, the continuous indicator variables representing the AUDIT score, given that it was above zero, were used (upper part of *Figure 2*). In this latter part of the model, substance non-use within each time point was treated as missing data, following standard assumptions of data missing at random (MAR; *Little and Rubin, 1987*). In that way participants who did not drink alcohol throughout the study contributed little information to the growth parameter estimates, but all information to alcohol use was used to estimate the growth parameters. We used Maximum Likelihood estimator for our analyses.

We controlled for the effects of age at baseline, sex, total brain volume (TBV) and site (by dummy coding eight of all nine neuroimaging sites, for simplicity we represent this by only one manifest variable in *Figure 2*) onto the slope of the continuous part of the growth model. As criteria for model fit we report Root Mean Square Error of Approximation (RMSEA), Comparative Fit Index (CFI), and Standardized Root Mean Square Residual (SRMR). Values of the CFI above 0.90 denote a well-fitting model, whereas for the RMSEA and the SRMR values less than 0.08 may be interpreted as acceptable model-fit.

Since our main interest was, which voxels of the brain from grey matter and white matter maps at age 14 years predict the continuous slope representing the increase of alcohol drinking over time, the final SEM models were conducted including a brain variable. For this purpose, we ran separate two-part growth mixture models for each and every voxel of the brain and repeated this analyses for the grey and white matter maps separately. This approach is exploratory as no specific areas of the brain are extracted (as would be the case in the more traditional analyses of ROI). While it bears the risk of inflated type-II-errors, it also enables the investigation of links between brain structures and behavior that have not been strictly established yet. For each SEM that was generated with a specific voxel as covariate, we plotted the estimate/p-value of the regression path from brain to the slope back into the brain, to be able to plot the resulting image using R (the scripts can be obtained from the first author, more flexible functions that may be helpful in conducting similar analyses can be found in the 'neuropointillist' R package, https://github.com/IBIC/neuropointillist, *Madhyastha et al., 2018*). This approach is not limited to growth curve models but can easily be extended to all types of models that can be implemented in MPlus). Finally, we thresholded the resulting maps with a threshold of p<0.001 and a cluster threshold of k > 100. We repeated the same analyses while predicting the discrete slope from voxels of the grey and white matter maps, although our main interest was on brain-wise predictors of the slope derived from the continuous part of the model.

## Acknowledgements

This work received support from the following sources: the European Union-funded FP6 Integrated Project IMAGEN (Reinforcement-related behaviour in normal brain function and psychopathology) (LSHM-CT- 2007–037286), the Horizon 2020 funded ERC Advanced Grant 'STRATIFY' (Brain network based stratification of reinforcement-related disorders) (695313), ERANID (Understanding the Interplay between Cultural, Biological and Subjective Factors in Drug Use Pathways) (PR-ST-0416–10004), BRIDGET (JPND: BRain Imaging, cognition Dementia and next generation GEnomics) (MR/N027558/1), the FP7 projects IMAGEMEND(602450; IMAging GEnetics for MENtal Disorders) and MATRICS (603016), the Innovative Medicine Initiative Project EU-AIMS (115300–2), the Medical Research Council Grant 'c-VEDA' (Consortium on Vulnerability to Externalizing Disorders and Addictions) (MR/N000390/1), the Swedish Research Council FORMAS, the Medical Research Council, the National Institute for Health Research (NIHR) Biomedical Research Centre at South London and Maudsley NHS Foundation Trust and King's College London, the Bundesministeriumfür Bildung und Forschung (BMBF grants 01GS08152; 01EV0711; eMED SysAlc01Z $\times$ 1311A; Forschungsnetz AERIAL 01EE1406A, 01EE1406B), the Deutsche Forschungsgemeinschaft (DFG grants SM 80/7–2, SFB 940/2), the Medical Research Foundation and Medical research council (grant MR/R00465X/1). Further support was provided by grants from: ANR (project AF12-NEUR0008-01 - WM2NA, and ANR-12-SAMA-0004), the Fondation de France, the Fondation pour la Recherche Médicale, the Mission Interministérielle de Lutte-contre-les-Drogues-et-les-Conduites-Addictives (MILDECA), the Assistance-Publique-Hôpitaux-de-Paris and INSERM (interface grant), Paris Sud University IDEX 2012; the National Institutes of Health, Science Foundation Ireland (16/ERCD/3797), U.S.A. (Axon, Testosterone and Mental Health during Adolescence; RO1 MH085772-01A1), and by NIH Consortium grant U54 EB020403, supported by a cross-NIH alliance that funds Big Data to Knowledge Centres of Excellence. SK has been funded by a Heisenberg grant from the German Science Foundation (DFG KU 3322/1–1, SFB936/C7), the European Union (ERC-2016-StG-Self-Control-677804) and the Jacobs Foundation (JRF 2016–2018). FN has been funded by a Heisenberg grant from the German Science Foundation (NE 1383/14–1).

## Additional information

### Competing interests

Christian Büchel: Reviewing editor, *eLife*. Tobias Banaschewski: Has served as an advisor or consultant to Bristol-Myers Squibb, Desitin Arzneimittel, Eli Lilly, Medice, Novartis, Pfizer, Shire, UCB, and Vifor Pharma; he has received conference attendance support, conference support, or speaking fees from Eli Lilly, Janssen McNeil, Medice, Novartis, Shire, and UCB; and he is involved in clinical trials conducted by Eli Lilly, Novartis, and Shire; the present work is unrelated to these relationships. The other authors declare that no competing interests exist.

### Funding

| Funder | Grant reference number | Author |
| --- | --- | --- |
| Horizon 2020 Framework Programme | | Simone Kühn |
| H2020 European Research Council | | Simone Kühn |
| Seventh Framework Programme | | Simone Kühn |
| Bundesministerium für Bildung und Forschung | | Simone Kühn |
| Max-Planck-Gesellschaft | Open-access funding | Simone Kühn |
| Jacobs Foundation | JRF 2016-2018 | Simone Kühn |

The funders had no role in study design, data collection and interpretation, or the decision to submit the work for publication.

## Author contributions
Simone Kühn, Software, Formal analysis, Funding acquisition, Visualization, Writing—original draft, Writing—review and editing; Anna Mascharek, Formal analysis, Writing—original draft, Writing—review and editing; Tobias Banaschewski, Arun Bodke, Uli Bromberg, Christian Büchel, Erin Burke Quinlan, Sylvane Desrivieres, Herta Flor, Antoine Grigis, Hugh Garavan, Penny A Gowland, Andreas Heinz, Bernd Ittermann, Jean-Luc Martinot, Frauke Nees, Dimitri Papadopoulos Orfanos, Luise Poustka, Sabina Millenet, Juliane H Fröhner, Michael N Smolka, Henrik Walter, Robert Whelan, Data curation, Writing—review and editing; Tomas Paus, Jürgen Gallinat, Data curation, Writing—original draft, Writing—review and editing; Gunter Schumann, Conceptualization, Data curation, Writing—review and editing; Ulman Lindenberger, Data curation, Formal analysis, Writing—review and editing

## Author ORCIDs
Simone Kühn (iD) https://orcid.org/0000-0001-6823-7969
Anna Mascharek (iD) https://orcid.org/0000-0001-7923-080X
Christian Büchel (iD) https://orcid.org/0000-0003-1965-906X
Dimitri Papadopoulos Orfanos (iD) https://orcid.org/0000-0002-1242-8990
Ulman Lindenberger (iD) https://orcid.org/0000-0001-8428-6453

## Ethics
Human subjects: Written informed consent was obtained from all participants as well as from their legal guardians. The adolescents were recruited from secondary schools. The study was approved by the local ethics committees (in Germany this was accomplished by the medical ethics committee of the University of Heidelberg, reference number: 2007-024N-MA) and approved by the head teachers of the respective schools.

## Decision letter and Author response
Decision letter https://doi.org/10.7554/eLife.44056.010
Author response https://doi.org/10.7554/eLife.44056.011

## Additional files
### Supplementary files
• Transparent reporting form
DOI: https://doi.org/10.7554/eLife.44056.008

### Data availability
This study uses human brain data which cannot be completely de-identified. Moreover, it was not part of the written consent of the participants for the data to be publicly shared. Researchers may access the dataset through a request to the IMAGEN consortium: https://imagen-europe.com/resources/imagen-project-proposal/.

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
