## [Decision Letter]

Thank you for submitting your article "Predicting development of adolescent drinking behaviour from whole brain structure at 14 years of age" for consideration by *eLife*. Your article has been reviewed by two peer reviewers, and the evaluation has been overseen by a Reviewing Editor and Michael Frank as the Senior Editor. The reviewers have opted to remain anonymous.

The reviewers have discussed the reviews with one another and the Reviewing Editor has drafted this decision to help you prepare a revised submission.

Summary:

In this manuscript, a study is reported in which structural MRI data of a large group of adolescents at age 14 is used to predict changes in future alcohol use. The authors use a novel voxel-wise VBM-based employment of structural equation modeling (SEM). Results show a significant change and interindividual heterogeneity in use versus non-use with age and significant interindividual heterogeneity but no change on average in drinking frequency. When relating these growth models to grey and white matter voxels at baseline, a positive association is found between grey matter clusters in the caudate nucleus bilaterally and the left cerebellum with changes in drinking frequency (but not with changes in alcohol use versus non-use).

Essential revisions:

1) It is not very clear from the manuscript what data were collected at each timepoint. It seems MRI data were only collected at time point 1 whereas AUDIT was collected at three timepoints. However, in various places the manuscript refers to longitudinal analysis of MRI data – which can easily be misinterpreted as suggesting that this is what was considered here – e.g., final line of Abstract: "The study is the first to demonstrate the feasibility of running separate voxel-wise structural equation models thereby opening new avenues for longitudinal data analysis in brain imaging". Please clarify. Also in Results – when the grey matter prediction is mentioned please clarify that this is for brain imaging at baseline/age 14. Similarly, in Discussion please specify that it is brain imaging at a single timepoint that is considered. Also, in the Discussion, where challenges of interpreting the findings are outlined, it could again be mentioned that from single time point imaging data it isn't possible to infer brain trajectories. Finally, "Although we focussed on grey matter volume the cerebellum is also present in our advanced analysis including considerably more participants and time points as well as less variability in terms of age at time of scanning." From this sentence it is not clear that there is brain data only at time-point 1.

2) The AUDIT data are not presented in either table form or as scatter or box plots. It is essential to see these data for each of the three test points. It is likely that the distributions are seriously skewed. Also essential is a count of how many youth meet criteria for Alcohol Use Disorder. How do the dichotomous use/non-use and continuous frequency of use factors map onto the actual data?

3) Similarly, the data for the MRI outcomes need to be presented. The path analyses of the SEMs lack meaning and preclude critiquing without knowledge about the underlying data and their distributions.

4) What is the nature of the growth trajectories? How did they serve in the prediction?

5) As stated above, one of the main assets of this manuscript is the use and description of the whole-brain voxel-wise employment of SEM. However, at some points the manuscript would benefit from a more detailed description of this method. Given the possibilities of this method for other researchers, the authors should release the code used for data analyses. Preferably with some (simulated) example data and a short tutorial.

6) Use of the AUDIT: total scores are used, whilst the AUDIT might measure both alcohol consumption and alcohol-related consequences (e.g., Doyle et al., 2007). Are different brain measures predicting (changes in) use and (changes in) consequences? This is important since the authors equate the total AUDIT with alcohol frequency throughout the manuscript.

---

## [Author Response]

Essential revisions:1) It is not very clear from the manuscript what data were collected at each timepoint. It seems MRI data were only collected at time point 1 whereas AUDIT was collected at three timepoints. However, in various places the manuscript refers to longitudinal analysis of MRI data – which can easily be misinterpreted as suggesting that this is what was considered here – e.g., final line of Abstract: "The study is the first to demonstrate the feasibility of running separate voxel-wise structural equation models thereby opening new avenues for longitudinal data analysis in brain imaging". Please clarify. Also in Results – when the grey matter prediction is mentioned please clarify that this is for brain imaging at baseline/age 14. Similarly, in Discussion please specify that it is brain imaging at a single timepoint that is considered. Also, in the Discussion, where challenges of interpreting the findings are outlined, it could again be mentioned that from single time point imaging data it isn't possible to infer brain trajectories. Finally, "Although we focussed on grey matter volume the cerebellum is also present in our advanced analysis including considerably more participants and time points as well as less variability in terms of age at time of scanning." From this sentence it is not clear that there is brain data only at time-point 1.

We are very grateful for this observation. In the new version of the manuscript we have included multiple statements that brain imaging was only considered at baseline/age 14 throughout the text to ensure that this fact is highlighted.

The sentence was confusing and therefore removed. Initially we wanted to highlight that in contrast to the cited paper that used FreeSurfer, our Voxel-based morphometry approach includes cerebellum in the same type of analysis. However we think this is not particularly relevant in this context.

2) The AUDIT data are not presented in either table form or as scatter or box plots. It is essential to see these data for each of the three test points. It is likely that the distributions are seriously skewed. Also essential is a count of how many youth meet criteria for Alcohol Use Disorder. How do the dichotomous use/non-use and continuous frequency of use factors map onto the actual data?

In order to illustrate the AUDIT data and how the dichotomous and frequency of use data maps onto the actual data the new version of the manuscript contains a new figure (Figure 1).

The reviewer is correct in pointing out that the AUDIT data is skewed. This is why we chose to use a statistical method that is designed to address zero inflated, skewed data, namely the two-part latent growth curve model. In order to ensure that this is sufficiently described in the manuscript we have added the following paragraph:

“Analyses were conducted within a structural equation modelling (SEM) framework using MPlus and R. […] In our study, the original distribution of the alcohol use variable (AUDIT-score) was decomposed into two parts (see Figure 1).”

In order to address the problem of skewness we additionally repeated the performed analyses with log-transformed data and observed very similar results. We now mention this in the Results section of the new version of the manuscript:

“We repeated the same analysis on log-transformed data, to address the problem of skewness in the data with almost identical results.”

We added Table 1 to illustrate how many youth meet criteria for Alcohol Use Disorder according to the WHO definition.

3) Similarly, the data for the MRI outcomes need to be presented. The path analyses of the SEMs lack meaning and preclude critiquing without knowledge about the underlying data and their distributions.

The findings that we present are the result of a whole-brain analysis. We ran a separate SEM model for each and every voxel in the brain. Therefore we have different estimates and p-values for the path between the latent slope and the “brain” variable for each separate model and therefore it is impossible to assign a single value to the path from the “brain” variable to the latent slope in what is now Figure 2. We thought that pictures displaying the clusters of significant results are helpful to illustrate the results (now Figure 3).

In order to illustrate the distributions of the MRI data we have plotted histograms of the mean within each significant cluster (x-axis in probability of grey matter units) (Author response image 1).

If the editor and reviewers find this instructive we could include this figure in the manuscript.

4) What is the nature of the growth trajectories? How did they serve in the prediction?

In order to describe the nature of the growth trajectories in more detail we added a paragraph with more detail in the Results section:

“Turning to the growth parameters, for the continuous part of the model the estimated mean of the slope was not significantly different from zero, indicating that, on average, no change in drinking habits emerged over time. […] No significant correlation emerged between intercept and slope for the continuous part of the model, indicating that a relation between alcohol use score at age 14 and change in this behaviour was not captured in our model.”

In order to better describe how the slope served in the prediction we added the following sentence in the Results section:

“Based on this two-part latent growth curve model on the clinical data we entered the brain data and conducted a whole-brain analysis on grey matter probability maps at age 14 years predicting change in alcohol use score, that is the latent slope in the continuous part of the model, over time from each voxel in the brain. Within the model an increase in alcohol use score is reflected as a positive and a decrease in alcohol use score as a negative latent slope score.”

5) As stated above, one of the main assets of this manuscript is the use and description of the whole-brain voxel-wise employment of SEM. However, at some points the manuscript would benefit from a more detailed description of this method. Given the possibilities of this method for other researchers, the authors should release the code used for data analyses. Preferably with some (simulated) example data and a short tutorial.

We have added some more detail on the description of the method. E.g. in the Introduction:

“The innovation of this paper’s approach lies in analysing the whole brain on voxel level. […] This approach offers the major advantage of giving consideration to both, specific structures of the brain on a rather fine-graded level, and the broad, exploratory search for generally influential areas.”

and in the Materials and methods section:

“This approach is exploratory as no specific areas of the brain are extracted (as would be the case in the more traditional analyses of ROI). […] This approach is not limited to growth curve models but can easily be extended to all types of models that can be implemented in MPlus, e.g. also measurement models.”

and in the Discussion section:

“To our knowledge up to now structural equation modelling on brain imaging data has been conducted solely based on data derived from regions of interest (ROIs)(Kievit et al., 2014; Kühn et al., 2017; Raz et al., 2005). […] This offers a new avenue of structural equation modelling on neuroimaging data in an unbiased, comprehensive way.”

At present our scripts are unfortunately not coded flexibly enough to be released as a toolbox. Therefore for now, we would like to refer interested readers to contact us to obtain the code. However, we are happy to refer readers to the neuropointillist R toolbox. This toolbox does offer the possibility of flexible data input and result output. It can likewise handle brain structural data.

Therefore we now refer to the neuropointillist R package in the new version of the manuscript, see citation above.

6) Use of the AUDIT: total scores are used, whilst the AUDIT might measure both alcohol consumption and alcohol-related consequences (e.g., Doyle et al., 2007). Are different brain measures predicting (changes in) use and (changes in) consequences? This is important since the authors equate the total AUDIT with alcohol frequency throughout the manuscript.

We are very grateful for this observation. Our decision to investigate the AUDIT total scores was an a priori choice and was based on the feature that it measures consumption and consequences at the same time. The fact that we did refer to the term “frequency” throughout the manuscript actually had a different reason. In previous papers describing two-part latent growth curve models the two model parts are commonly referred to as the “probability” part of the model and the “frequency” part of the model (see e.g. Vazsonyi and Keiley, 2007). In order to avoid the confusion with the frequency subscale of the AUDIT and to be more precise in the terminology we now refer to the “alcohol use score”.

However, we used the time to additionally compute the separate analyses on grey matter volume for the frequency subscale of the AUDIT score.

Audit frequency subscore, continuous model, slope as predictor: n.s.

continuous model, intercept as predictor: n.s.

dichotomous model, slope as predictor: n.s.

dichotomous model, slope as predictor: n.s.

And found no significant results. Which made it even more relevant to remove the “frequency” terminology throughout the manuscript.

In order to avoid a major multiple test problem due to the fact that we would have to compute and report 8 separate analyses (grey vs. white matter X dichotomous vs. continuous X slope vs. intercept) for each subscore of the Audit (frequency, symptoms, problems) which would result in 24 additional tests we would like to refrain from conducting and reporting all of these subanalyses.